# Can octopus embryos and juveniles contend with heatwaves?

Jorge Arturo Vargas-Abúndez[1], Ana Karen Meza-Buendia[1], Olivia Alvarado[2],
Sharon Valdez-Carbajal[2], Maite Mascaró[1], Claudia Caamal-Monsreal[1],
J. Alejandro Kurczyn-Robledo[3], Gabriela Rodríguez-Fuentes[4], Carlos Rosas[1]*

1 Unidad Multidisciplinaria de Docencia e Investigación, Facultad de Ciencias, Universidad Nacional Autónoma de México. Puerto de abrigo s/n, Sisal, Yucatán, México, 2 Posgrado en Ciencias del Mar y Limnología, Universidad Nacional Autónoma de México, Facultad de Ciencias. Puerto de abrigo s/n, Sisal, Yucatán, México, 3 Laboratorio de Ingeniería y Procesos Costeros, Instituto de Ingeniería, Universidad Nacional Autónoma de México, Sisal, Yucatán, México, 4 Unidad de Química en Sisal, Facultad de Química, Universidad Nacional Autónoma de México, Sisal, Yucatán, México

* crv@ciencias.unam.mx

## Abstract

Heatwaves are emerging climatological threats intensifying by climate change, that pose unprecedented challenges to thermally sensitive marine species. This study investigated the physiological and metabolic responses of *O. maya* offspring to heatwave conditions, focusing on oxidative stress, mitochondrial function, and survival. We simulated a critical scenario where females with an optimal thermal history (24°C) laid eggs at the onset of a heatwave, exposing the offspring to optimal (24°C), intermediate (26°C), or high (30°C) temperatures for the entire embryonic development (~45 days) and 30 days post-hatching. Embryos incubated at 30°C showed altered morphometry (reduced mantle and arm lengths) and suppressed routine metabolic rates by the end of embryonic development. Among antioxidants analyzed, total glutathione (GSH) emerged as a key factor in mitigating oxidative stress, supporting previous observations suggesting a key role in reactive oxygen species (ROS) protection. We hypothesized that energy reallocation to stress defense mechanisms compromised developmental processes, resulting in smaller hatchlings with reduced survival and diminished factorial metabolic scope. High-resolution respirometry revealed mitochondrial dysfunction, including increased proton leak and reduced respiratory efficiency, exacerbating oxidative damage and impairing oxygen transport. While some juveniles exhibited metabolic plasticity and elevated ATP production, these responses were insufficient to counteract the long-term costs of thermal stress. These findings suggest that although optimal thermal history, as seen in upwelling zones, may offer temporary protection, prolonged exposure to elevated temperatures could severely compromise reproductive success and population sustainability.

**Data availability statement:** Raw data are available at https://doi.org/10.5281/zenodo.14026237.

**Funding:** This study was partially financed by the Universidad Nacional Autónoma de México (UNAM) through its Programa de Apoyo a Proyectos de Investigación e Innovación Tecnológica [CR IN203022] and to Dirección General de Asuntos del Personal Académico (DGAPA) for Posdoctoral fellowship to Jorge Arturo Vargas Abúndez, and Consejo Nacional de Humanidades, Ciencias y Tecnología (CONAHCYT) PRONAII-2024-70 grant to CR. Open Access funding provided by Universidad Nacional Autonoma de Mexico. The funders had no role in study design, data collection and analysis, decision to publish, or preparation of the manuscript.

**Competing interests:** The authors have declared that no competing interests exist.

## 1. Introduction

Heatwaves are discrete, prolonged warming events that occur in both the atmosphere and oceans [1]. Marine heatwaves are defined as periods during which sea surface temperatures (SST) exceed the 90th-percentile threshold for at least five consecutive days, resulting in extended thermal anomalies [2]. With continued climate change, these events are expected to increase in both intensity and frequency, posing potentially severe and poorly understood consequences for terrestrial and marine animals [3]. In marine ecosystems, heatwaves can significantly affect species abundance, particularly when they coincide with critical periods such as reproduction [4,5].

Temperature exerts a governing role in all biochemical processes, elevating metabolic rates, and thereby physiological energy production [6]. As metabolic rates increase, harmful free radicals are produced as metabolic by-products due to increased electron flux in the mitochondrial electron transport system. If these free radicals are not neutralized by the antioxidant system, they can lead to oxidative stress [7–10].

Mitochondrial ROS play a dual role in cellular processes, functioning as crucial signaling molecules at low concentrations to regulate oxidative metabolism, cellular differentiation, and autophagy, promoting cell survival under stress conditions [11]. However, when ROS levels surpass the antioxidant defense capacity, excessive oxidative stress leads to significant cellular damage and can ultimately result in cell death [11].

The red octopus, *Octopus maya*, has served as a research model to address the effects of temperature on ectothermic marine organisms [12–16]. This cephalopod is particularly vulnerable to high temperatures [17]. Its optimal breeding temperature range narrowly between 24°C and 25 °C [13,18]; above 26°C disruption in energy balance occur in both sexes, likely due to compromised oxygen transport efficiency, which affects cellular respiration and reproduction [8,19].

*O. maya* females exhibit the remarkable ability to finely control reproduction timing, optimizing conditions for their progeny to thrive under favorable thermal conditions. Transcriptomic studies indicate that upregulation of genes of the amide family, and other genes, in the optical gland (located in the octopus brain), are implicated in the control of spawning [20,21]. Myomodulin and APGW-amide genes are involved in egg-laying regulation and the onset of egg fertilization [21,22]. Likewise, it was observed that genes associated with fertilization like LGALS3, VWC2, and Pcsk1 were downregulated at elevated temperatures, reducing the union between spermatozoon and the extracellular matrix of the egg, altogether explaining why high temperatures could be inhibiting *O. maya* spawning [21].

Thermally, the Yucatan Peninsula, Mexico, where *O. maya* inhabits, presents an interesting dynamic in relation to octopus reproduction and associated fauna. This marine region experiences seasonal upwellings that stabilize coastal water temperatures and provide critical habitats for reproduction and development. In this region, seasonal upwelling takes place between March and November when the Yucatan current strengthens and pushes waters from the sub-surficial layer (about

250m deep) from the Caribbean Sea towards the Gulf of Mexico [23,24], keeping the Yucatan continental shelf at about 23–28°C through the year [25]. Upwelling regions have been shown to offer refuge for thermally sensitive species, including *O. maya* [26].

Unfortunately, the intensity of the Caribbean current, which feeds the Yucatán Peninsula's continental shelf, may be diminishing due to environmental changes driven by climate change [27]. This would produce annual temperature increases around 0.0161°C yr-1, which means 1.6°C in 100 years [27].

Fishery studies suggest that preadults and adults of *O. maya* remain in the deep zones of the shelf, feeding and growing until the reproductive season starts when males and females migrate to the coastal zone to meet and spawn [28]. Avendaño et al. [28] registered temperatures around 24°C at 35-40m deep in the Yucatán continental shelf where pre adults and adults of *O. maya* were also sampled, suggesting that 35-40m deep could be the thermal refuge zone for the growth of late juveniles and preadults before they migrate to the coastal zone for reproduction. In the coast, we have recorded bottom temperatures (10.5m depth) surpassing 28°C for periods as long as three months (Fig 1). In addition, in 2024, two heatwaves were detected in the *O. maya* distribution area with records of SST exceeding 28°C: one from May 16 to June 02, and another from August 3–10. These coincided with the expected migration of females from deep waters to coastal areas for copulation during vitellogenesis [29]. If this kind of thermal anomaly continues, or worse, increase in intensity and frequency, as they are expected [30,31], they will affect different life cycle moments, modifying reproductive conditions, embryo development, and conditioning juvenile survival.

In this study, we explored a critical scenario where females experienced an optimal temperature (24°C), as expected under thermal protection provided by Yucatan upwelling, but lay eggs during an intense heatwave (reaching temperatures of 30°C). Our goal was to assess whether the progeny could endure such thermal conditions and to understand the metabolic alterations, protective mechanisms, and potential consequences on embryonic development, growth, and hatchling

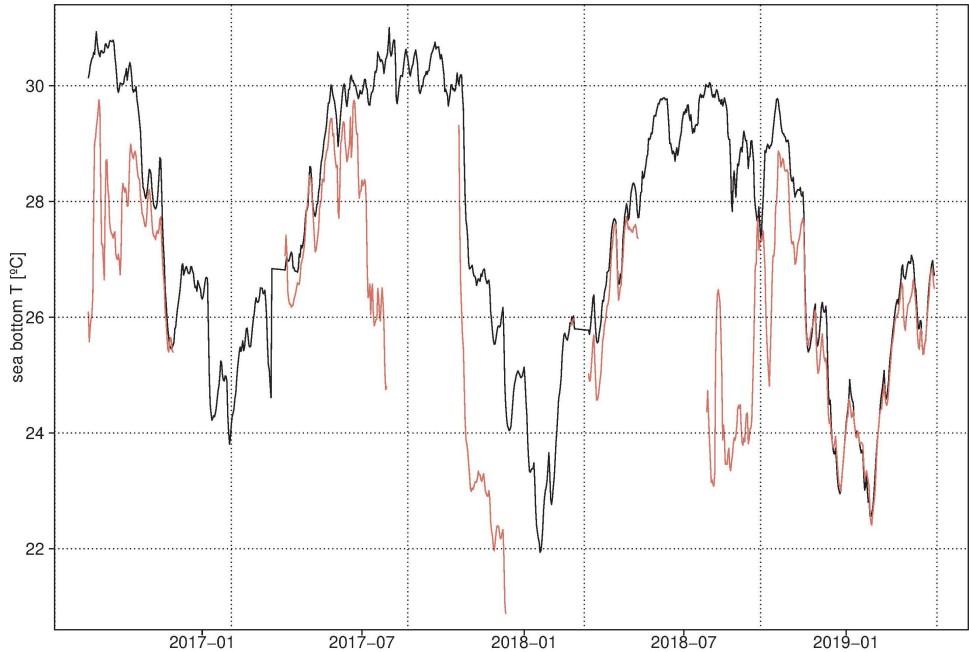

**Fig 1. Sea bottom temperatures at Campeche (19.97°N, 90.87°W; black line) and Sisal (21.28°N, 90.05°W; red line) during the period 2016–2019.** Daily average temperatures were derived from hourly time series recorded by thermometers installed on two Acoustic Doppler Current Profilers (ADCPs).

viability. We hypothesized that prolonged exposure to elevated temperatures (30 °C) will compromise developmental processes and long-term viability of *O. maya* offspring through (i) energy reallocation toward stress defense at the expense of growth, (ii) impaired mitochondrial efficiency and altered metabolic rates, and (iii) reduced survival in juveniles. To test these predictions, we tracked individuals through their entire ~45-day embryonic period and the first 30 days after hatching. We integrated a morphometric analysis with key oxidative-stress markers and metabolic rates profiles. In addition, and for the first time in juvenile octopus, we performed high-resolution respirometry on isolated mitochondria to further understand at the subcellular level the link between mitochondrial dysfunction and long-term heat stress. This approach provides a comprehensive understanding on the mechanisms involved in the *O. maya* heat-stress response, and shed light on the species' potential vulnerability and adaptive capacity under climate change scenarios.

## 2. Materials and methods

### Ethics statement

In this study, octopuses were anesthetized with 3% ethanol in seawater at experimental temperatures [32,33] to facilitate humane euthanasia [34], in accordance with ethical protocols [35] and with consideration for the animals' welfare during manipulations [36,37]. This approach took into account the nociception of aquatic invertebrates [38]. The study adhered to the European directive on the use of cephalopods in research (DIRECTIVE 2010/63/UE) and was approved by the Bioethical Commission at the Faculty of Sciences, National Autonomous University of Mexico (UNAM) (CEARC/Bioética/25102021). Furthermore, the research complied with Mexican legal standards for the humane killing of animals (NOM-033-SAG/ZOO-2014, derogating to NOM-033-ZOO-1995) and followed the principles outlined in the Five R's for invertebrates [39].

### 2.1. Origin of animals

A group of ten *O. maya* males and ten females sexually mature (400–700 g) were captured in the Sisal coast off the Yucatan Peninsula (21°9′55″N, 90°1′50″W) by a local drift-fishing method known as 'Gareteo'. The specimens were caught during one collection trip in March 2022 and transferred to outdoor 6-m-diameter flow-through systems provided with shade mesh, protein skimmers and 500-µm bag filters. Conditioning lasted 10 days ($35 \pm 1$ salinity; dissolved oxygen (DO) > 5.5 mg L$-$ 1; $28 \pm 1$ °C) with a density of 1 animal m$^{-2}$ and 2-PVC open tubes per animal as refuge. Octopuses were fed twice a day (0900 and 1700 h) with a semi-moist paste made with squid and crab meat, gelatin, a vitamin–mineral premix and ascorbic acid, at a ratio of 8% of its wet weight (WW) [40]. Daily monitoring of the animals' general health was performed through careful inspection of their behavior and feeding activity.

After conditioning, three sexually mature and fertilized females were randomly selected and placed in 80-L individual indoor tanks at 24°C (Fig 2). Temperature of 24°C was controlled with room air conditioning. Females were maintained in experimental conditions for 20 days and fed following the same feeding protocol of the conditioning period. A fiberglass box per tank was placed as a refuge and spawn settlement. Each tank was connected to a semi-closed recirculation system coupled with a rapid-rate sand filter. Water parameters other than temperature showed similar values as conditioning period; pH was kept above 8 and photoperiod at 12L/12D with low light intensity (30 Lux m$^{-2}$).

Once the spawning started, eggs of each female were obtained after eight days, just when spawning ended. Subsequently, 600 eggs per spawn (1200 eggs total) were randomly sampled and artificially incubated at optimal (24°C), intermediate (26°C) or high temperature (30ºC; Fig 2) till hatching (30–40 days). Daily cleaning and maintenance included siphoning out detritus and performing visual inspections of the overall health of the eggs. Since female octopuses stop feeding at the onset of spawning and eventually die naturally, they were euthanized using a humane protocol described for juveniles [41] and approved by the bioethical guidelines of the Faculty of Sciences at the National Autonomous University of Mexico (UNAM). To minimize distress and prevent pain, the specimens were anesthetized with 3% ethanol

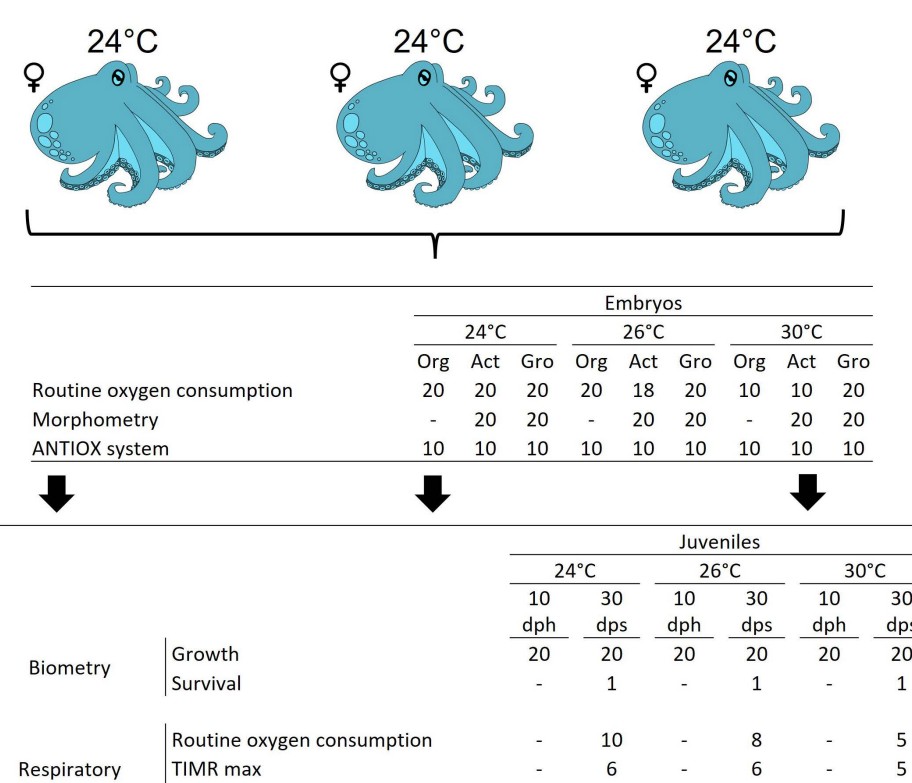

| | Embryos | | | | | | | | |
|---|---|---|---|---|---|---|---|---|---|
| | 24°C | | | 26°C | | | 30°C | | |
| | Org | Act | Gro | Org | Act | Gro | Org | Act | Gro |
| Routine oxygen consumption | 20 | 20 | 20 | 20 | 18 | 20 | 10 | 10 | 20 |
| Morphometry | - | 20 | 20 | - | 20 | 20 | - | 20 | 20 |
| ANTIOX system | 10 | 10 | 10 | 10 | 10 | 10 | 10 | 10 | 10 |

| | | Juveniles | | | | | |
|---|---|---|---|---|---|---|---|
| | | 24°C | | 26°C | | 30°C | |
| | | 10 dph | 30 dps | 10 dph | 30 dps | 10 dph | 30 dps |
| Biometry | Growth | 20 | 20 | 20 | 20 | 20 | 20 |
| | Survival | - | 1 | - | 1 | - | 1 |
| Respiratory metabolism | Routine oxygen consumption | - | 10 | - | 8 | - | 5 |
| | TIMR max | - | 6 | - | 6 | - | 5 |
| | TIMR min | - | 6 | - | 6 | - | 5 |
| | TMS | - | 1 | - | 1 | - | 1 |
| Mitochondrial metabolism | Respiratory state 3 | - | 2 | - | 3 | - | 2 |
| | Respiratory state 4 | - | 2 | - | 3 | - | 2 |
| | OXPHOS capacity | - | 2 | - | 3 | - | 2 |
| | Respiratory control | - | 2 | - | 3 | - | 2 |
| ANTIOX system | | - | 10 | - | 10 | - | 10 |

**Fig 2. Experimental design and number of replicates for each experimental group.** TIMR = Temperature-Induced Metabolic Rate (maximum: max; minimum: min); TMS = Thermal Metabolic Scope; ANTIOX = Antioxidant Defense System; dph = days post-hatching; dps = days post-spawning. Org, Act, and Gro represent the organogenesis, activation, and growth stages of embryo development, respectively. Females were maintained at 24°C, and their embryos were incubated at either 24°C, 26°C, or 30°C throughout development. After hatching, juveniles remained at the same temperature they experienced as embryos. The downward arrows indicate that all juveniles originated from females kept at 24°C and continued in their respective temperature treatments (24°C, 26°C, or 30°C) after hatching.

dissolved in seawater for an average of 12–15 minutes. Once a lack of response to external stimuli was observed, the brain carefully desensitized with a deep incision between the eyes.

## 2.2. Experiments with embryos

### 2.2.1. Routine metabolic rate (RMR).
Every five days, respiratory metabolism was measured individually (n = 10–20; Fig 2): embryos were placed in micro-plate clear glass vials with integrated sensor spots (750 µL volume, Loligo Systems, Copenhagen, DK) and maintained in seawater in experimental incubation temperature (Fig 2). Simultaneously, oxygen consumption of five control chambers (vials without embryo) were also measured. Prior to each trial, the flow-through cell

O$_2$ sensor was calibrated at 24°C, 26°C or 30°C, according to the embryo incubation temperature, using air-saturated seawater (100%) and with 1–1.5% anhydrous sodium sulphite (at 0%). Vials were submerged in a transparent glass container with temperature-controlled seawater maintained at each experimental temperature. The container was placed on a Sensor Dish Reader (Presens, Regensburg, DE) that recorded oxygen concentration measurements of 24 vials simultaneously every 15 s. The measurement time was adjusted according to the oxygen concentration in vials, which were never lower than 80% of the saturation level. This time was around 30 minutes. All measurements were graphed according to time and embryo oxygen consumption (MO$_2$) was calculated as follows:

$$MO_2 = \left( \frac{O_{2(A)} - O_{2(B)}}{\Delta t} \right) \times \frac{V}{M}$$

where MO$_2$ is respiration rate (mg O$_2$ g WW$^{-1}$ h$^{-1}$); O$_{2(A)}$ is initial oxygen concentration in the chamber (mg O$_2$ L$^{-1}$); O$_{2(B)}$ is final oxygen concentration in the chamber (mg O$_2$ L$^{-1}$); V is water volume in the chamber minus water volume displaced by the embryo (expressed as L); t is time elapsed during measurement (h); and M is egg body mass containing the embryo (mg WW).

**2.2.2. Embryonic development.** Immediately after respiration rate measurements, eggs, and their corresponding embryos were photographed (Fig 2), maintaining eggs at their corresponding experimental temperature (24°C, 26°C or 30°C). The morphological parameters of *O. maya* eggs' eye diameter (ED, mm), arm length (AL, mm); mantle length (ML, mm), total length (TL, mm), yolk volume (YV, mm$^3$) and wet weight (WW, g) were measured. Each egg was observed and photographed using Leica EZ4 HD (Wetzlar, DE) stereoscopic microscope whose software (Leica LAS EZ, Wetzlar, DE) allows identification of embryonic stage and size. The developmental stages were identified [42,43](Caamal-Monsreal et al. 2016; Naef 1928) and grouped as organogenesis (when possible), activation (when the heart starts its activity; stages XV–XVI; n = 20), and growth (stages XVII–XIX; n = 20) following the criteria obtained in previos base line studies [44]. Afterward, embryos were weighted (WW; ± 0.001 g) and immediately stored in liquid nitrogen on Eppendorf tubes, and stored at −80º C until analysis.

**2.2.3. Antioxidant defense (ANTIOX) system.** The frozen embryos, at organogenesis, activation and growth stages, were individually homogenized in cold buffer 0.05 M Tris pH 7.4 at 100 mg tissue/mL using a Potter–Elvehjem homogenizer (30 embryos per experimental temperature and 10 per stage of development, 90 in total, Fig 2). Homogenate samples used for superoxide dismutase (SOD), catalase (CAT), and glutathione-s-transferase (GST) were centrifuged at 10,000 g for 5 min at 4 °C, and the supernatant was separated for analysis. All samples were stored at −80 °C until analysis; all assays were done in duplicates. A Sigma-Aldrich assay kit (19,160) (USA) was used to evaluate SOD, using Dojindo's highly water-soluble tetrazolium salt, WST-1(2-(4-Iodophenyl)-3-(4-nitrophenyl)-5-(2,4-disulfophenyl)- 2H tetrazolium, monosodium salt) that produces a water-soluble formazan dye upon reduction with a superoxide anion. The reduction rate with O$_2$ is linearly related to the xanthine oxidase (XO) activity and inhibited by SOD. CAT activity was measured according to Góth [45] modified by Hadwan and Abed [46]. In this method, undecomposed H$_2$O$_2$ is measured with ammonium molybdate after three minutes to produce a yellowish color with maximum absorbance at 374 nm. Total glutathione (GSH) was measured with Sigma- Aldrich Glutathione Assay Kit (CS0260) (St. Louis MO, US). This kit utilizes an enzymatic recycling method with glutathione reductase [47]. The GHS sulfhydryl group reacts with Ellman's reagent and produces a yellow-colored compound read at 405 nm. The GST activity was determined from the reaction between reduced glutathione and 1-chloro-2.4-dinitrobenzene at 340 nm [48]. Proteins were analyzed in supernatant according to Bradford [49] and used to normalize enzyme activities. To evaluate oxidative damage caused by radical oxygen species (ROS), carbonyl groups in oxidized proteins (OP) and lipid peroxidation (LPO) were measured in the sampled embryos. PO was estimated following the 2,4-dinitrophenylhydrazine alkaline protocols developed by Mesquita et al. [50] and reported in nmol/mg wet weight. For this assay, 200 µl of 2,4 dinitrophenylhydrazine (10 mM in 0.5 M HCL) were

incubated with 200 μl of the sample homogenate and 100 μl of NaOH (6 M). Absorbance was read at 450 nm after 10 min of incubation at room temperature against a blank where an equal volume of homogenization buffer substitutes the protein solution. LPO was evaluated using Peroxi Detect Kit (PD1, Sigma-Aldrich, USA) following the manufacturer's instructions. In this assay, peroxide oxidizes $Fe^{2+}$ ions at acidic pH, forming a colored adduct with xylenol orange that is measured at 560 nm.

## 2.3. Experiments with juveniles

### 2.3.1. Experimental system set up.
To assess the effect of long-term water temperature exposure beyond embryo development, embryos from the different temperature treatments were individualized upon hatching in plastic containers (1.41-L) submerged in 50-L tanks (18H × 72L × 45W cm), with five to ten containers per tank (90 animals in total, 30 per temperature treatment), following the same temperature of origin (24°C, 26°C or 30°C) for 30 days (Fig 2). Tanks of the same temperature treatment were connected to a 300-L semi-closed recirculatory seawater system equipped with mechanical and UV filtration. Octopus containers were provided with a PVC tube as a refuge, which had two openings covered with plastic mesh (2 mm) that prevented juveniles from escaping. Air stones and water recirculation in tanks provided current for proper water oxygenation of each container.

Water temperature was controlled with CW1000 aquarium chillers and 1800-Watt titanium heaters. Juvenile octopuses were individually fed twice a day (09:00 and 17:00 h) with a heat-dried (40 °C) pelleted diet made with the same ingredients and proportions as the adult diets, at a ratio of 8% of its body weight [40]. Every day, feces and uneaten food were siphoned out before feeding, and visual inspections of the juveniles' overall health were conducted based on their behavior and feeding activity. Juvenile survival was recorded daily. Despite our efforts, we have not identified specific criteria that unequivocally predicts mortality, which makes it challenging to prevent cephalopod mortality in our study. Wet weight was assessed by measuring (precision 0.001 g, balance model PMB 53, Adam Equipment, USA) of randomly selected individuals at 10 and 30 days post hatch (n = 20, N = 120; Fig 2).

Daily growth coefficient (DGC) was calculated as:

$$DGC = ((\ln Wf - \ln Wi)/t) \times 100$$

where Wf is the final wet weight, Wi, the initial wet weight, and t, the number of days (20).

### 2.3.2. Juvenile respiratory metabolism. 2.3.2.1. *Routine metabolic rate (RMR).*
RMR of juveniles was measured individually in closed respirometric chambers (17 mL) at the corresponding acclimation temperature (24°C, 26°C or 30 °C) at the end of the experiment (30 days post hatch). A different number of juvenile replicates were analyzed per experimental temperature, 10, 8 and 5 for 24°C, 26°C and 30°C, respectively, due to the differences of survival registered between experimental temperatures (Fig 2). A similar respirometric protocol as for the embryos respirometric measurements was followed, except that dissolved oxygen measurements were recorded in this case every second using flow-through oxygen sensors (Loligo systems, Denmark) connected by an optical fiber to Witrox 4 amplifiers (Loligo systems, Denmark) and using an optical oxygen sensor (Loligo systems Denmark) placed at the internal glass side of the respirometric chamber to allow following the oxygen concentration variations during measurements.

**2.3.2.2. *High metabolic rate (HMR) and low metabolic rate (LMR).*** Following a standardized method to induce HMR and LMR [19,51,52], oxygen consumption was measured when animals were exposed to temperature that induced maximum metabolic rate (TIMR max), defined as 90% of critical thermal maxima (CTMax), and minimum metabolic rate (TIMR min) as 110% of critical thermal minima (CTMin).

Thus, considering CTMax and CTMin data by Noyola et al. [13,14], TIMR max was run at 30, 31 and 33 °C for animals raised at 24°C, 26°C and 30°C, respectively, whereas TIMR min at 14, 15 and 21°C for animals raised at 24°C, 26°C

and 30°C respectively. For each trial, juveniles were randomly selected from their acclimation tank and rapidly placed in a closed respirometric chamber submerged in a temperature-controlled seawater bath maintained at the corresponding TIMR max or TIMR min temperature for 5–6 min or 30 min, for TIMR max and TIMR min, respectively. This time is long enough for the animals to display its maximum and minimum metabolic rates, without the activation of compensatory mechanisms [51]. A control chamber without animals was run simultaneously for background respiration correction. Different groups of juveniles (5–6 per experimental condition; Fig 2) were exposed to TIMR max or TIMR min to avoid any physiological interference with the measurement due to previous high or low temperature exposure. TIMR max and TIMR min were recorded as previously described for RMR.

Thermal metabolic scope (TMS) and factorial metabolic scope (FMS) were then calculated as:

$$TMS = TIMR\ max - TIMR\ min,$$

$$FMS = TIMR\ max / TIMR\ min,$$

where TIMR max = temperature-induced metabolic rate maximum (mean value), and TIMR min = temperature-induced metabolic rate minimum (mean value) (both as mg $O_2$ $g^{-1}$ $h^{-1}$).

After routine oxygen consumption, TIMR max and TIMR min measurements, the WW (g) of each juvenile was obtained (± 0.001 g; Fig 2), and immediately stored in liquid nitrogen on Eppendorf tubes, and stored at −80ºC until analysis.

**2.3.3. Mitochondrial respiratory metabolism.** 2.3.3.1. *Mitochondrial isolation*. Functional mitochondria were isolated from randomly selected juveniles collected at the end of the experiment (Fig 2), adapting the protocol by Meza-Buendia *et al*. [53]. Pools of two to five juveniles were used per sample (n = 3 for 24°C and 26°C, and n = 2 for 30°C). Each organism was weighed and subsequently anesthetized by leaving them in cold seawater (11°C) until they presented a state of relaxation evidenced by a decrease in their heart rate and loss of muscle tone: flaccidity and immobility of the mantle and arms, reduced adhesiveness of the suckers and absence of response to mechanical stimuli. Animals were then euthanized by placing them in a glass Petri dish on ice. They were then homogenized using a Potter-Elvehjem PTFE mortar and glass tube (Sigma- Aldrich P7859-1EA) operated by a drill at 500 rpm with 2 mL of isolation buffer (500 mM sucrose, 300 mM KCl, 2 mM ethylene glycol-bis (β-aminoethyl)-N, N, N′, N′-tetraacetic acid (EGTA), 25 mM 4-(2-hydroxyethyl)- piperazine-1-ethanesulfonic acid (HEPES), 1.5% w/v fatty acid-free bovine serum albumin (FA-free BSA), pH 7.4 at 20°C, 826 mOsM). To keep mitochondria alive and functional the whole isolation process was carried out on ice (4°C). The homogenate was centrifuged at 3,528 rcf for 5 minutes (4°C), and then the supernatant recovered and centrifuged at 7,939 rcf (4°C) for 15 minutes. The supernatant was subsequently discarded by decantation and 2 mL of isolation buffer without FA-free BSA was added to re-suspend the pellet formed. Finally, the sample was centrifuged at 7,939 rcf for 15 min, the resulting supernatant discarded, and the formed pellet re-suspended with 60 µl of isolation buffer without FA-free BSA. An aliquot (5–10 µl) of the mitochondrial isolation sample was used to estimate the concentration of mitochondrial proteins [49]. The rest of the sample was kept cold (4°C) for use in high resolution respirometry within the first 4 hours [53].

2.3.3.2. *Mitochondrial O2 Consumption Measurements*. *In vitro* mitochondrial oxygen consumption was determined according to Meza-Buendia *et al*. [53], using an Oxygraph-2k™ (O2k, Oroboros Instruments, Austria), which consists of an enclosed respirometer with two 2 mL chambers using a polarographic oxygen sensor to detect oxygen ($O_2$) flux of ± 1 pmol $O_2$-$s^{-1}$-$mL^{-1}$. To start up the trial, the oxygen sensors of the Oxygraph-2k™ were calibrated at the same juvenile acclimation temperature (24°C, 26°C, or 30°C), using a MiR05 respiration medium, consisting of (in mM): 0.5 EGTA, 3 $MgCl_2$, 60 lactobionic acids, 20 Taurine, 10 $KH_2PO_4$, 20 HEPES, 110 D-Sucrose and 1 g/L FA-free BSA (pH 7).

Once the Oxygraph-2kTM system reached the steady state of basal oxygen consumption of the working system, where the rate of oxygen consumption is constant, isolated mitochondria (0.3 mg) were added and a series of substrates and

inhibitors added as well to characterize mitochondrial capacity. Specifically, proline (Pro) was added as a mitochondrial exogenous substrate, at a final concentration of 5 mM, to avoid depolarization of the mitochondrial membrane potential. Then ADP (1.25 mM) was added to induce oxidative phosphorylation (OXPHOS) respiration state (State 3′, S3′) and Oligomycin (0.0025 mM) to induce the oligomycin-induced LEAK state (State 4′o, S4′o). Finally, rotenone (0.0025 mM) plus antimycin A (12.5 µM) were added to obtain the residual respiration ROX, which corresponds to non-mitochondrial respiration.

**2.3.3.3. *Mitochondrial parameters*.** The different respiratory rates measured in the respiratory states were used to calculate mitochondrial parameters. The mitochondrial respiratory states S3′ (S3 = S3′-ROX) and S4′o (S4 = S4′o-ROX) were corrected for the respiratory state ROX. After correction, the respiratory control ratio (mitochondrial coupling; RCR) was defined as S3/S4o, while the mitochondrial respiratory rate attributed to OXPHOS (OXPHOS capacity, using proline as exogenous substrate) was defined as S3–S4o. All reported values were normalized by total protein content (pmol $O_2$ $s^{-1}$ mg protein$^{-1}$) except for the respiratory control ratio.

**2.3.4. Antioxidant defense (ANTIOX) system.** The frozen juveniles were processed following the same protocol described for embryos (10 juveniles per experimental temperature, 30 in total; Fig 2).

## 2.4. Statistical analysis

Changes in embryo RMR were assessed by two-way analysis of variance (ANOVA), with temperature (24°C, 26°C, and 30°C) and stage of development (organogenesis, activation and growth) as fixed factors. Changes in DGC, RMR, TIMR max, TIMR min, and mitochondrial parameters (State 3 and 4, OXPHOS capacity, and RCR) of juveniles were assessed via one-way analysis of variance (ANOVA), with temperature (24°C, 26°C, and 30°C) as the fixed factor. Normality and homogeneity of variance were verified using residual plots, the Shapiro-Wilk test, and F-test. Data not meeting these assumptions (embryo RMR, mitochondrial respiratory state 4 and respiratory control) were transformed using natural logarithmic transformations. Tukey's post hoc test was applied when ANOVA significance was observed.

Morphometric changes and antioxidant system responses were evaluated using Principal Coordinate Analysis (PCO) followed by permutational multivariate analysis of variance (PERMANOVA) [54]. PCO was employed to visualize variation patterns among embryos and juveniles across different temperature groups. In each PCO, a representation of the relative distances and position of within-group centroids was embedded to aid interpretation.

Resemblance matrices were constructed for each developmental stage (organogenesis, activation, growth and juvenile stage) using Euclidean distances between sample pairs. Raw data were pre-treated with natural logarithm (Log [X + 1]) and z-normalized. The statistical significance of PCO-derived groups was assessed using one-way PERMANOVA, with temperature as a fixed explanatory factor. Significance testing involved 999 unrestricted permutations of raw data and residuals under the reduced model, generating empirical distributions of pseudo-F values [55]. Post hoc comparisons were conducted for significant main tests (P < 0.05), comparing centroids of identified groups. All univariate tests and visualizations were performed using R software version 4.2.2 [56] or Prism5 (GraphPad Software). Multivariate analyses were conducted using PRIMER 6 and PERMANOVA+ for PRIMER. Data are presented as mean ± SD. Raw data are available at https://doi.org/10.5281/zenodo.14026237.

## 3. Results

### 3.1. Embryo morphometry

PCOs of the morphological characteristics (WW, TL, ML, AL, ED, YV) of *O. maya* embryos explained 70.3% (Fig 3A) and 68.9% (Fig 3B) of the total variation in the dataset, namely activation and growth stage, respectively. (S1 Table).

At the activation stage, the PCO showed a clear temperature-dependent separation of embryo morphometry (Fig 3A). This separation occurred primarily along the horizontal axis, with embryo samples at 24°C, 26°C, and 30°C positioned on the left, center, and right sides of the PCO, respectively.

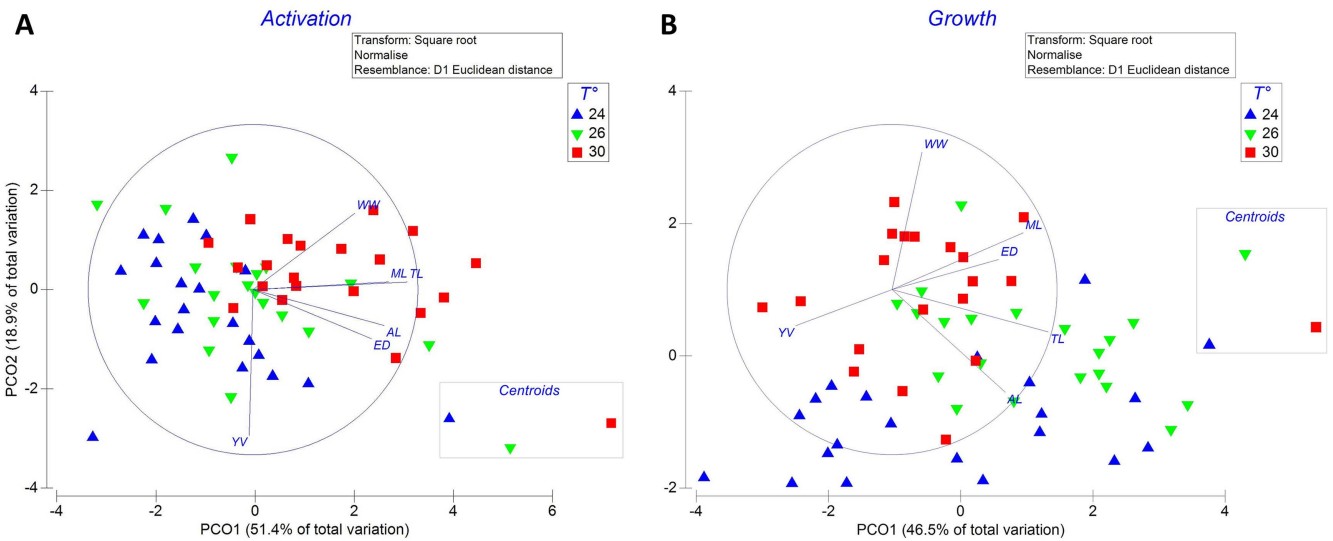

**Fig 3. Morphological characteristics of *Octopus maya* embryos incubated at 24°C, 26°C, or 30°C.** Panels show data for the activation (A) and growth (B) stages of development. WW = Wet weight; TL = Total length; ML = Mantle length; AL = Arm length; ED = Eye diameter; YV = Yolk volume. Individual data points are shown for each treatment.

Eigenanalysis revealed that the left side of the ordination was inversely correlated with all morphometric measurements, indicating that lower exposure temperatures corresponded to smaller embryo lengths and sizes. Conversely, higher temperatures were associated with increased embryo dimensions. Embryo lengths and sizes were negatively correlated with yolk volume.

PERMANOVA results supported the significant effects of temperature on embryo morphometry (pseudo-F = 6.0, P = 0.001, 999 unique permutations; S1 Table).

At the growth stage, embryos incubated at 30°C displayed the highest wet weight, along with increased mantle length and eye diameter (Fig 3B). However, these embryos also showed reduced yolk volume, and total length and arm length compared to those incubated at 24°C and 26°C (S1 Table).

These relative reductions in total length and arm length as temperature increased showed significant differences (pseudo-F = 5.2, P = 0.001, 999 unique permutations; S1 Table).

### 3.2. Embryo respiratory metabolism

A two-way ANOVA revealed a significant interaction between developmental stage and incubation temperature ($F_{(interaction)}$ = 11.96, P < 0.001, Partial η² = 0.24 [$IC_{95=}$ 0.14–1.00]), indicating that the effect of temperature on routine metabolic rate (RMR) differed across stages (Fig 4). At organogenesis, Tukey's post hoc tests showed that embryos at 26 °C had significantly higher RMR than those at both 24 °C and 30 °C (P < 0.05, comparisons versus 24 °C and 30 °C). At activation, no significant differences in RMR were detected among the three temperatures (all P > 0.05). At growth, RMR at 24 °C was significantly greater than at 26 °C (P < 0.05, comparison versus 26 °C), with 30 °C falling intermediate but not differing significantly from either group (Fig 4).

### 3.3. Embryo ANTIOX defense system

PCOs revealed temperature-dependent clustering in the antioxidant defense profiles of *O. maya* embryos across development (Fig 5). PERMANOVA confirmed significant thermal modulation during organogenesis (pseudo-F = 4.8, P = 0.001) and

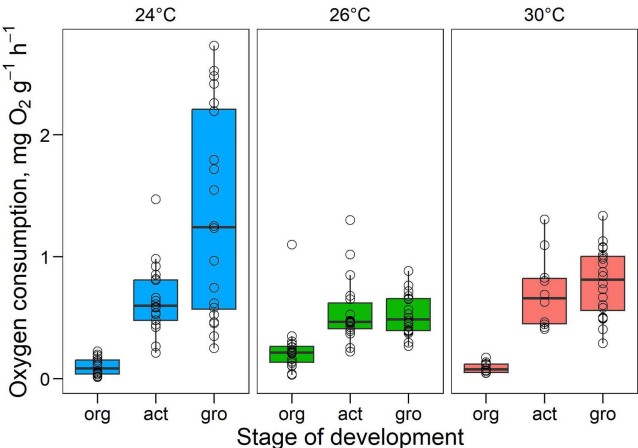

**Fig 4. Routine oxygen consumption rates (RMR) of *Octopus maya* embryos incubated at 24°C, 26°C, and 30°C.** Values are expressed as mg O₂ g⁻¹ h⁻¹. Org, Act, and Gro correspond to the organogenesis, activation, and growth stages of development, respectively. Data are shown as mean±SD, with individual data points included.

activation (pseudo-F=4.0, *P*=0.001; 999 permutations each). Notably, clusters at these stages showed no vectorial alignment with oxidative stress biomarkers (LPO, PO) or enzymatic antioxidants (SOD, GSH and GST), indicating absent heat-induced oxidative damage or compensatory enzymatic induction. In contrast, growth-stage embryos exhibited divergent thermal responses: specimens incubated at 30°C formed a discrete cluster in the upper ordination quadrant with pronounced convergence toward the GSH vector (pseudo-F=2.5, *P*=0.01; 999 permutations). This trajectory implicates glutathione metabolism as a predominant compensatory mechanism supporting embryonic thermotolerance during advanced development.

### 3.4. Juvenile growth and survival

Water temperature significantly influenced the juvenile DGC, with higher temperatures promoting increased DGC (F=6.6, P<0.003). Specifically, DGC values were 1.83±1.37 at 24°C, 2.75±1.24 at 26°C, and 3.37±1.42 at 30°C. In contrast, survival rates exhibited an inverse relationship with temperature, decreasing markedly as temperatures rose. Survival rates were 92.5%, 82.5%, and 52.5% for juveniles incubated at 24°C, 26°C, and 30°C, respectively.

### 3.5. Juvenile respiratory metabolism

Water temperature significantly influenced juvenile respiratory metabolism, specifically RMR and TIMR min, but not TIMR max (Fig 6A, B and C). In routine (Fig 6A), juveniles reared at 26°C and 30°C exhibited significantly higher oxygen consumption rates (0.7±0.1 and 0.6±0.2 mg O₂ g⁻¹ h⁻¹, respectively) compared to those at 24°C (0.3±0.2 mg O₂ g⁻¹ h⁻¹) (F=13.8, P<0.001, η²=0.58 [IC$_{95=}$0.30–1.00]). For TIMR min (Fig 6B), juveniles at 30°C showed significantly higher oxygen consumption (0.2 mg O₂ g⁻¹ h⁻¹) than those at 24°C and 26°C (0.08±0.01 and 0.1±0.0 mg O₂ g⁻¹ h⁻¹, respectively; F=4.01, P<0.05, η²=0.36 [IC$_{95=}$0.01–1.00]). Regarding TIMR max (Fig 6C), no statistically significant differences were observed (F=1.2, P=0.32, η²=0.15 [IC$_{95=}$0.00–1.00]). Although the TMS among temperature groups was similar, FMS revealed a clear trend. Juveniles at 30°C exhibited approximately two- to three-fold reductions in FMR values compared to 24°C and 26 °C, respectively (Fig 6D).

### 3.6. Juvenile mitochondrial respiratory metabolism

High-resolution respirometry of isolated mitochondria from *O. maya* juveniles revealed no statistically significant changes (F=4.1, P=0.08, η²=0.92 [IC$_{95=}$0.55–1.00]) in State 3 respiration across experimental temperatures (Fig 7A). In contrast,

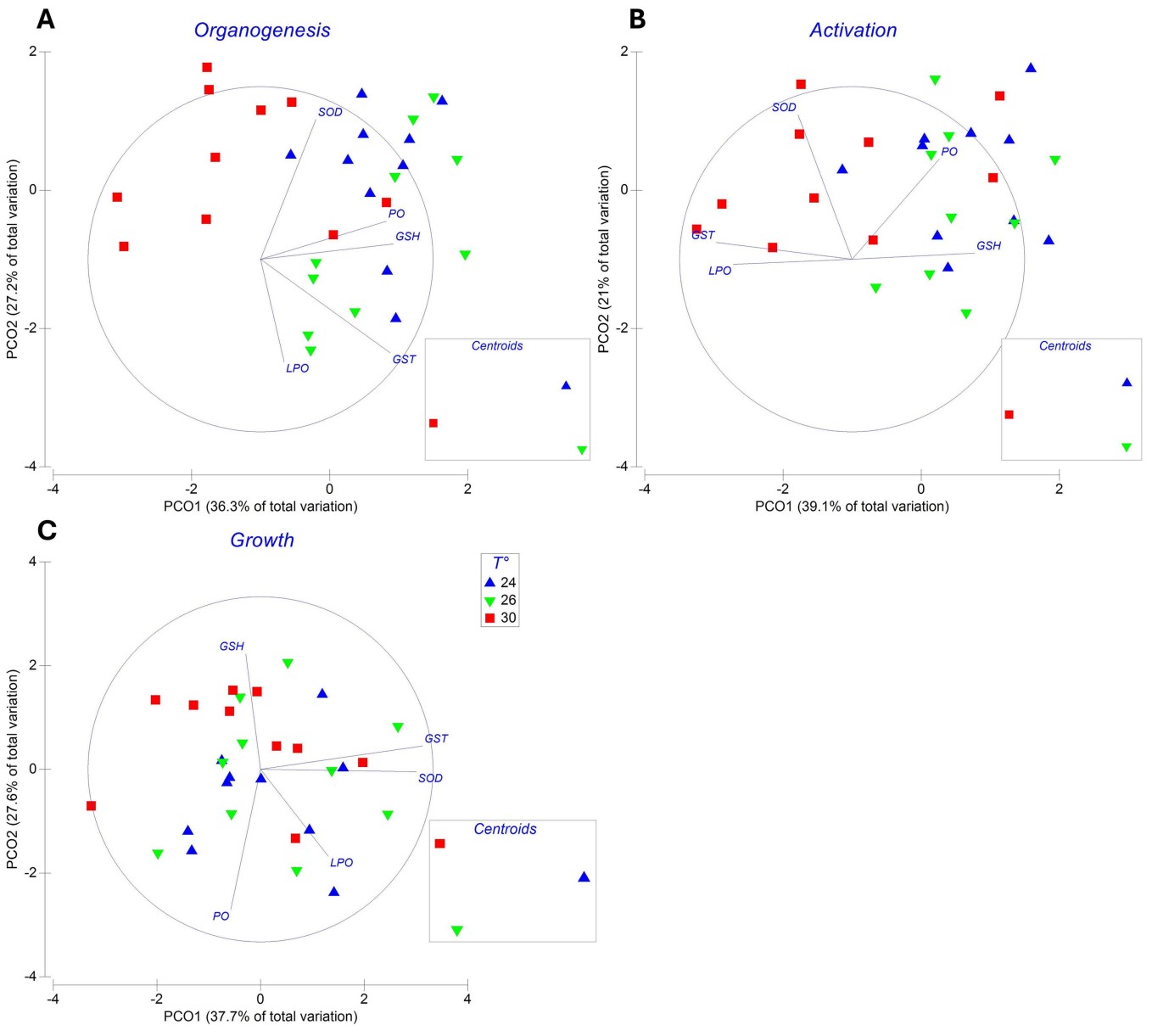

**Fig 5. Antioxidant enzyme activity and oxidative damage markers in *Octopus maya* embryos incubated at 24°C, 26°C, and 30°C.** Antioxidant components include the enzyme superoxide dismutase (SOD), the tripeptide total glutathione (GSH), and the enzyme glutathione-S-transferase (GST). Oxidative damage markers include lipid peroxidation (LPO) and oxidized proteins (PO). Panels represent the organogenesis **(A)**, activation **(B)**, and growth **(C)** stages of development. Each panel includes a 2-D ordination plot showing the relative distances and group centroids. Individual data points are shown for each treatment.

State 4'o respiration showed a progressive and significant increase with rising temperatures (F = 8.2, P = 0.03; η² = 0.56 [IC$_{95=}$0.00–1.00]; Fig 7B). OXPHOS capacity (Fig 7C) also increased with temperature, with mitochondria of juveniles acclimated to 30°C exhibiting significantly higher oxygen consumption rates compared to those at 26°C (F = 6.2, P < 0.05, η² = 0.85 [IC$_{95=}$0.26–1.00]). Conversely, RCR (Fig 7D) was highest in juveniles at 24°C (4.1 ± 0.1), followed by those at

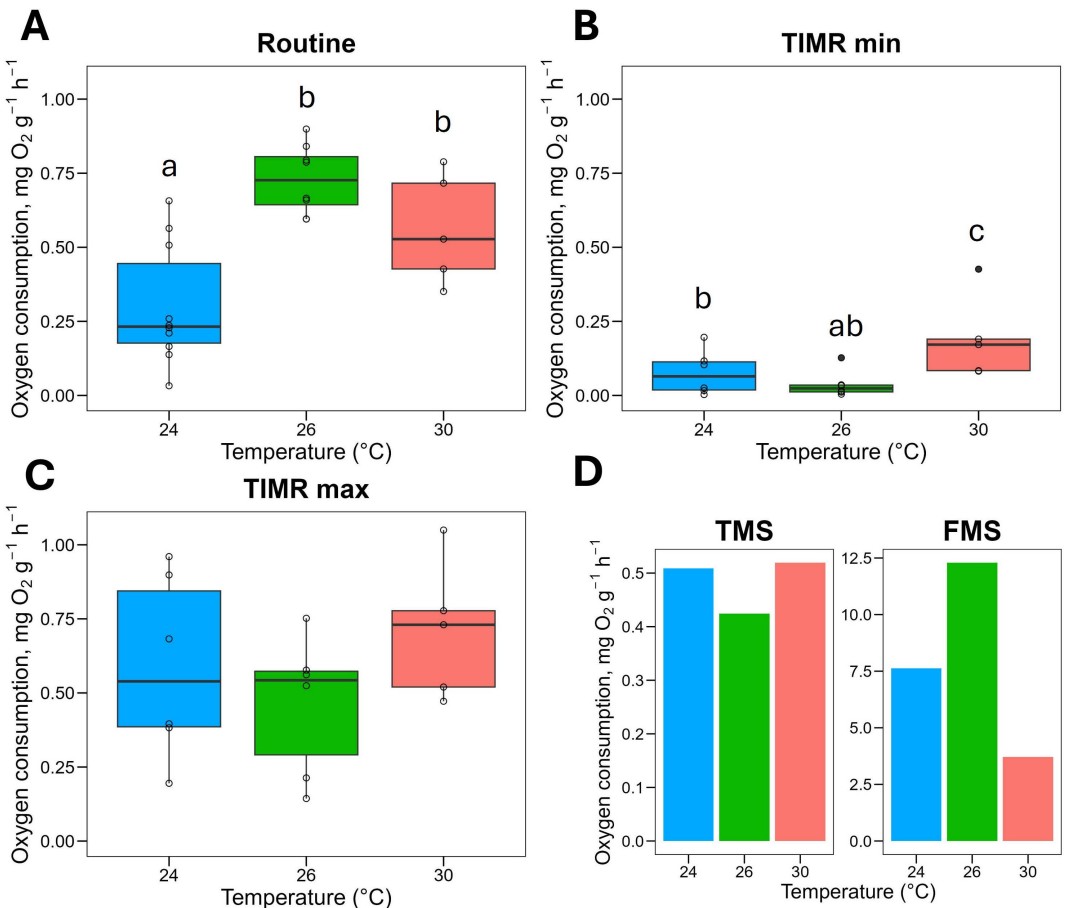

**Fig 6. Respiratory metabolism of juvenile *Octopus maya* chronically exposed to 24°C, 26°C, and 30°C, with thermal stress histories matching their embryonic incubation temperatures. (A)** Routine metabolic rate (RMR); (B) temperature-induced metabolic rate minimum (TIMR min); (C) temperature-induced metabolic rate maximum (TIMR max); (D) thermal metabolic scope (TMS) and factorial metabolic scope (FMS). Data are presented as mean ± SD, with individual data points shown for each group. Different lowercase letters indicate statistically significant differences among groups (P < 0.05).

30°C (3.1 ± 1.2) and 26°C (2.2 ± 0.3). However, these differences in RCR were not statistically significant (F = 4.2; P = 0.07 η² = 0.74 [IC$_{95=}$0.00–1.00]).

### 3.7. Juvenile ANTIOX defense system

The antioxidant defense system of *O. maya* juveniles displayed distinct patterns depending on acclimation temperature, as evidenced by the PCO (Fig 8). The clustering of juvenile samples revealed clear groupings based on temperature, reflecting differences in antioxidant enzyme activity profiles. Juveniles acclimated to 24°C were primarily located on the right side of the ordination plot, closely aligned with vectors representing SOD, CAT, and GST enzyme activity. This alignment indicates higher activity levels of these enzymes and suggests a key role in shaping the antioxidant profile at this temperature. In contrast, juveniles at 30°C clustered on the lower-left side of the ordination plot, characterized by high levels of GSH and PO values but lower lipid LPO. Juveniles acclimated to 26°C displayed a more dispersed distribution, reflecting a heterogeneous antioxidant response across individuals. These findings were statistically supported by

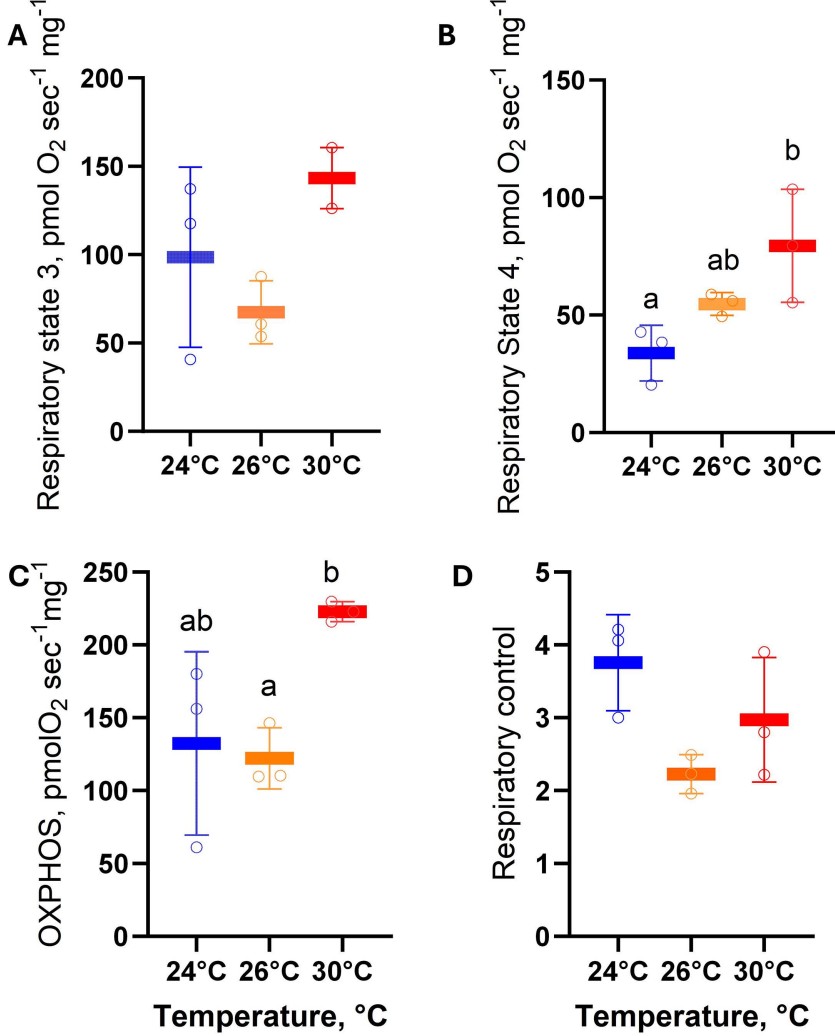

**Fig 7. Mitochondrial respiratory metabolism of juvenile *Octopus maya* chronically exposed to 24°C, 26°C, and 30°C, with thermal stress histories matching their embryonic incubation temperatures.** Parameters evaluated include State 3 respiration **(A)**, State 4'o respiration **(B)**, oxidative phosphorylation (OXPHOS) capacity **(C)**, and the respiratory control ratio (RCR; **D)**. State 3: Oxygen consumption in the presence of exogenous substrates (proline) and ADP, representing active mitochondrial respiration. State 4'o: Oligomycin-induced oxygen consumption, reflecting proton leakage and electron transport chain activity. OXPHOS capacity: Calculated as State 3 – State 4'o, representing ATP synthase activity. RCR: The ratio of State 3 to State 4'o, indicating mitochondrial coupling efficiency. Data are presented as mean ± SD, with raw data points represented as colored circles. Different lowercase letters denote statistically significant differences (P < 0.05).

PERMANOVA, which revealed significant differences among the temperature groups (pseudo-F = 2.1; P = 0.03; 998 unique permutations).

## 4. Discussion

Results obtained in the present study suggest that when non-thermally stressed octopus spawn during a simulated heatwave (30°C), embryos and subsequent juveniles can endure these thermal anomalies—but not without physiological costs. The early effects of the simulated heatwave conditions induced morphological alterations in embryos, including increased wet weight but reduced mantle and arm lengths compared to embryos incubated at 24°C. This suggests that

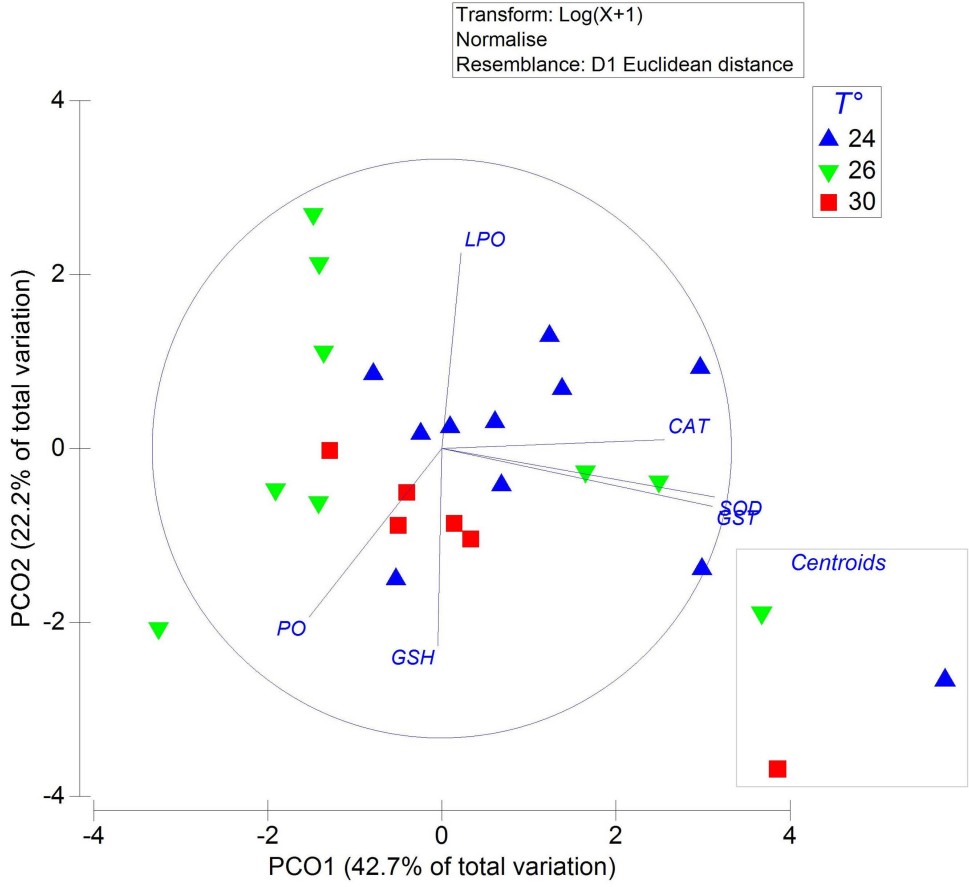

**Fig 8. Antioxidant enzyme activity and oxidative damage markers in juvenile *Octopus maya* chronically exposed to 24°C, 26°C, and 30°C, with thermal stress histories matching their embryonic incubation temperatures.** Antioxidant enzymes include superoxide dismutase (SOD), catalase (CAT), the tripeptide total glutathione (GSH), and glutathione-s-transferase (GST). Oxidative damage markers include lipid peroxidation (LPO) and oxidized proteins (PO). The embedded 2-D ordination plot shows the relative distances and centroids for each treatment group, highlighting clustering patterns and variability in antioxidant responses. Individual data points are shown for each treatment.

the higher living weight observed in heatwave-exposed embryos may result from water accumulation in the egg rather than biomass growth during development. This interpretation aligns with findings in *O. maya* and *O. vulgaris* where embryos of the former exhibited greater final sizes at optimal temperatures (100 mg ww at 22°C) versus sublethal heat stress (62 mg ww at 30°C) [42,57]. Transcriptomic evidence further confirms downregulation of genes governing development/morphogenesis, neurogenesis, and neuroendocrine differentiation under thermal stress [44], with altered expression of autophagy regulator ambra1a emerging as a primary candidate mechanism for the observed morphological trade-offs.

Heatwave-exposed embryos, as well as those exposed to moderate temperatures (26°C), exhibited a lower RMR during the growth phase. A similar result was observed by Caamal-Monsreal et al. [42], where incubation at 30°C, compared to 18°C, 22°C, and 26°C, reduced the RMR of *O. maya* embryos in advanced developmental stages. Additionally, hatching rates, wet weight, and size at hatching decreased markedly at 30°C. Metabolic alterations in *O. maya* embryos have been associated with a defective optic gland, likely resulting from the downregulation of several genes involved in homeotic transformation [44]. Notably, several studies have demonstrated a high degree of plasticity in the aerobic metabolism of *O. maya* embryos and juveniles [18,42,52,58]. The observed reduction in metabolic rate implies a decreased

availability of physiological energy for essential biological processes, such as growth, compared to embryos maintained at optimal temperatures (24°C). Similarly, in Atlantic salmon (*Salmo salar*) gametes, elevated temperatures enhanced thermal plasticity, but reduced hatching success, indicating high sensitive to heat stress, especially in sperm cells [59]. These findings suggest that embryos allocate energy predominantly toward heat stress defense and repair mechanisms under thermal stress conditions [60].

Multivariate analyses revealed that GSH activity during the growth stage was a key differentiating factor in heatwave-exposed embryos compared to those incubated at 24°C or 26°C, highlighting the special role of GSH in these conditions. GSH is a versatile tripeptide that plays auxiliary roles in the antioxidant defense system and other redox processes [61]. It acts as a cofactor for enzymes such as glutaredoxin (GRx), glutathione peroxidase (GPx), and GST, which facilitate the glutathione redox cycle using NADPH from the glucose metabolism pathway [62].

Interestingly, other key antioxidant enzymes, including SOD and GST, did not exhibit significant differences among the temperature groups. This suggests that metabolic energy in heatwave-exposed embryos may have been redirected primarily toward the antioxidant defense mechanisms and related molecules to mitigate oxidative stress.

Recent findings suggest that females maintained at 24°C transfer GSH molecules to their embryos via yolk vesicles, equipping the progeny to counteract oxidative damage during development [60]. This suggests a special role of GSH as a key antioxidant *in O. maya*'s stress response. During heatwave conditions, which likely result in elevated production of ROS, embryos appear compelled to synthesize GSH and other intermediates to maintain redox balance. As the most critical non-enzymatic antioxidant, GSH's thiol group effectively neutralizes ROS, preserving cellular homeostasis and mitigating oxidative damage [9].

In *O. maya*, as in other cephalopods, lipids are particularly sensitive to ROS attack. Uncontrolled levels of ROS can initiate a cascade of oxidative processes, targeting lipids in the yolk and cellular membranes. This oxidative damage compromises membrane integrity and yolk quality, ultimately leading to developmental malformations in embryos [57,63,64]. These findings underscore the dual importance of maternal GSH provisioning and embryonic antioxidant responses in safeguarding against lipid peroxidation and ensuring normal embryonic development under thermal stress conditions.

The possible reallocation of physiological energy from growth towards antioxidant defense mechanisms would help thrive under heatwave conditions but, do these smaller hatchlings have reduced capacity to tolerate subsequent thermal stress? Addressing this question through present longer-term study revealed that the physiological condition of *O. maya* hatchlings with a heat-stress thermal history was significantly compromised when exposed to high temperatures for an additional 30 days. Nearly half (48.5%) of the hatchlings chronically maintained at 30°C succumbed during this period, compared to 92.5% survival at 24°C. Survivals maintained at 30°C had highest growth rates, a moderate RMR, a TMS comparable to that of hatchlings maintained at 24°C, and the highest mitochondrial ATP production. These findings may lead to think that a subset of thermally stressed embryos and hatchlings developed physiological adjustments that could enabled them to endure and function under extreme thermal conditions.

However, previous studies have demonstrated that *O. maya* juveniles exhibits pronounced sensitivity to high temperatures, particularly when exposed to a constant 30°C for 20 days [13,14]. Hatchlings derived from non-thermally stressed females exhibited lower growth rates but achieved 100% survival [52,63]. Conversely, hatchlings from thermally stressed females showed markedly reduced survival, elevated metabolic rates, and heightened oxidative stress. These findings underscore the significant role of thermal history, particularly maternal inheritance, in exacerbating the offspring's sensitivity to high temperatures [52,63].

Findings from the present study suggest that metabolic depression is among the physiological mechanisms *O. maya* employs to recover from heat stress and tolerate subsequent high temperatures. At 30°C, the reduced RMR and TMS exhibited by the embryos, suggest prioritization of energy conservation over growth, which enabled them to maintain energy balance comparable to embryos maintained at 26°C. Following this period, juveniles displayed a re-acceleration of metabolic rates, demonstrating notable metabolic plasticity post-stress.

While many taxa exhibit the ability to depress metabolism, this trait is typically associated with species inhabiting extreme environments that commonly or seasonally experience extreme desiccation, anoxia, or temperatures extremes, such as *Artemia*, intertidal bivalves and snails [65–67]. However, in the case of octopuses, there are no obvious physiological or behavioral adaptations associated with extreme environments, indicative of a marked ability to depress metabolism. Nevertheless, as stated above, the evidence suggest that *O. maya* has a clear but limited ability to depress metabolism, mainly when the thermal stress was experienced since embryo stages. In the recent study by Vargas-Abúndez et al. [52], juveniles maintained under optimal temperature conditions (25°C) but derived from thermally stressed females significantly reduced their metabolic rate during the first eight days after hatching. This response likely reflects a mechanism to mitigate metabolic damage caused by ROS inherited from the progenitors.

Factorial aerobic scope (FAS, calculated as maximum metabolic rate/standard metabolic rate) is a key indicator of an animal's oxygen transport capacity relative to its baseline oxygen uptake rate [68]. In this study, we used TMS as a proxy for aerobic scope and FMS as a proxy for FAS. Our findings revealed that FMS in juveniles maintained at 30°C was 60% and 51% lower than those maintained at 26°C and 24°C, respectively indicating that the aerobic scope is strongly reduced at 30°C.

Results from high-resolution respirometry of isolated mitochondria suggested that the metabolic state of juveniles with a history of chronic thermal stress, which seemed to have succesfull adaptive responses, however, is closely associated with diminished oxygen transport capacity, increased proton leak, and reduced respiratory efficiency.

Consistent with the *in vivo* respirometry results, temperature significantly influenced metabolic function, with net maximal mitochondrial oxygen consumption (State 3) being elevated at 30°C relative to 24°C and 26°C. However, oxygen consumption associated solely with maintaining the proton gradient in response to "leakage" (State 4′o) exhibited a strong, positive relationship with temperature, highlighting a progressive decline in mitochondrial respiration efficiency with increasing temperature.

Similar findings have been reported in *O. maya* adults, where exposure to 30°C reduced heart mitochondrial metabolism, decreased ATP production, and impaired oxygen transport [8]. Although specific evaluations of these mechanisms in juveniles are lacking, it is plausible that similar processes occur. In juveniles exposed to 30°C, high temperatures may also reduce ATP production in heart mitochondria specifically, ultimately compromising oxygen transport and triggering metabolic depression as a compensatory strategy. While the mitochondrial metabolism measured in this study reflects the entire organism rather than being heart-specific, the data indicate that elevated temperatures increase mitochondrial membrane permeability, as evidenced by higher proton leak. This heightened proton leak likely reduces mitochondrial efficiency, further supporting the hypothesis that compromised oxygen transport and metabolic depression are key physiological responses to prolonged thermal stress.

Consistent with the mitochondrial metabolism results, the present study revealed elevated levels of PO in the whole body of juveniles maintained at 30°C. These findings were accompanied by decreased activities of SOD and CAT, indicating an imbalance in the antioxidant defense mechanisms critical for maintaining homeostasis. In *O. maya* juveniles, the high levels of proton leak observed under heatwave conditions may contribute to the oxidative damage detected, potentially compromising the survival of individuals in this experimental group, similar to reported in embryos (Repolho et al., 2014) and in wild *O. vulgaris* pre-adults exposed to polluted marine environments [69].

The present study highlights the physiological vulnerability of *O. maya* to elevated temperatures, underscoring the species' limited capacity to cope with chronic thermal stress. Despite its tropical distribution, *O. maya* thrives due to the cooling influence of the Yucatan upwelling, which mitigates regional temperatures that would otherwise surpass the species' thermal tolerance, as seen in areas less influenced by the upwelling. However, anticipated shifts in upwelling dynamics and rising surface temperatures, exacerbated by climate change, are expected to disrupt these favorable thermal environments. This disruption could lead to significant alterations in wild *O. maya* populations, potentially driving conservation challenges, forced migration, or local extinctions, with cascading ecological and socioeconomic consequences

for ecosystems and costal communities reliant on this species. Crucially, our findings emphasize that even moderately elevated temperatures, such as 26°C, produce measurable negative impacts, especially under prolonged thermal anomalies. These results serve as a warning of the potential risks posed by future ocean warming scenarios and highlight the urgency for conservation strategies to mitigate the impacts of climate change on thermally sensitive marine species.

## 5. Conslusion

The results of this study suggest that *O. maya* with an optimal maternal temperature has the capacity to withstand high temperatures (30°C) during embryo development by regulating antioxidant defense mechanisms to mitigate excessive oxidative damage. This regulation, potentially codified at the genetic level, incurs a cost, as evidenced by the production of smaller and less hatchlings compared to those incubated at 24°C or 26°C. While not all hatchlings from embryos incubated at 30°C survived subsequent exposure to the same temperature, the survivors demonstrated physiological strategies, including metabolic depression, enabling them to sustain life for at least the first 30 days post hatching.

These findings indicate that during intense and prolonged heatwaves, where embryos are exposed to temperatures as high as 30°C, a portion of the population may survive by seeking refuge in cooler, deeper waters. This underscores the need for further studies to investigate potential migrations during heatwaves and highlights the importance of regulating deeper-depth fisheries to safeguard these potential thermal refuges.

## Supporting information

**S1 Table. Statistical results and descriptive data supporting the findings of this study.**
(XLSX)

## Acknowledgments

Authors thank Omar Dominguez-Castanedo for help in experimentation of octopus embryos.

## Author contributions

**Data curation:** Jorge Arturo Vargas-Abúndez, Carlos Rosas.

**Formal analysis:** Jorge Arturo Vargas-Abúndez, Carlos Rosas.

**Funding acquisition:** Maite Mascaró, Carlos Rosas.

**Investigation:** Ana Karen Meza-Buendia, Olivia Alvarado, Sharon Valdez-Carbajal, Carlos Rosas.

**Methodology:** Sharon Valdez-Carbajal, Maite Mascaró, Carlos Rosas.

**Resources:** Claudia Caamal-Monsreal, Gabriela Rodríguez-Fuentes, Carlos Rosas.

**Supervision:** Jorge Arturo Vargas-Abúndez, Carlos Rosas.

**Validation:** Ana Karen Meza-Buendia, Maite Mascaró, Gabriela Rodríguez-Fuentes.

**Visualization:** Jorge Arturo Vargas-Abúndez, J. Alejandro Kurczyn-Robledo, Carlos Rosas.

**Writing – original draft:** Jorge Arturo Vargas-Abúndez, Sharon Valdez-Carbajal, Maite Mascaró, Carlos Rosas.

**Writing – review & editing:** Ana Karen Meza-Buendia, Olivia Alvarado, Claudia Caamal-Monsreal, J. Alejandro Kurczyn-Robledo, Gabriela Rodríguez-Fuentes, Carlos Rosas.

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
