## [Decision Letter · Decision Letter 0]

11 Jun 2025

PONE-D-25-14615Can octopus embryos and juveniles contend with heatwaves?PLOS ONE?

Dear Dr. Rosas,

Thank you for submitting your manuscript to PLOS ONE. After careful consideration, we feel that it has merit but does not fully meet PLOS ONE’s publication criteria as it currently stands. Therefore, we invite you to submit a revised version of the manuscript that addresses the points raised during the review process.

We look forward to receiving your revised manuscript.

Kind regards,

Juan J Loor

Academic Editor

PLOS ONE

**Journal Requirements:**

1. When submitting your revision, we need you to address these additional requirements. Please ensure that your manuscript meets PLOS ONE's style requirements, including those for file naming. The PLOS ONE style templates can be found at https://journals.plos.org/plosone/s/file?id=wjVg/PLOSOne_formatting_sample_main_body.pdf and https://journals.plos.org/plosone/s/file?id=ba62/PLOSOne_formatting_sample_title_authors_affiliations.pdf 2. Thank you for stating the following financial disclosure: This study was partially financed by the Universidad Nacional Autónoma de México (UNAM) through its Programa de Apoyo a Proyectos de Investigación e Innovación Tecnológica [CR IN203022] and to Dirección General de Asuntos del Personal Académico (DGAPA) for Posdoctoral fellowship to Jorge Arturo Vargas Abúndez, and Consejo Nacional de Humanidades, Ciencias y Tecnología (CONAHCYT) PRONAII-2024-70 grant to CR. Open Access funding provided by Universidad Nacional Autonoma de Mexico.   Please state what role the funders took in the study.  If the funders had no role, please state: "The funders had no role in study design, data collection and analysis, decision to publish, or preparation of the manuscript." If this statement is not correct you must amend it as needed. Please include this amended Role of Funder statement in your cover letter; we will change the online submission form on your behalf. 3. Please note that your Data Availability Statement is currently missing the repository name. If your manuscript is accepted for publication, you will be asked to provide these details on a very short timeline. We therefore suggest that you provide this information now, though we will not hold up the peer review process if you are unable. 4. Please include captions for your Supporting Information files at the end of your manuscript, and update any in-text citations to match accordingly. Please see our Supporting Information guidelines for more information: http://journals.plos.org/plosone/s/supporting-information.

Reviewers' comments:

Reviewer's Responses to Questions

**Comments to the Author**

1. Is the manuscript technically sound, and do the data support the conclusions?

Reviewer #1: Partly

Reviewer #2: Yes

2. Has the statistical analysis been performed appropriately and rigorously?

Reviewer #1: Yes

Reviewer #2: Yes

3. Have the authors made all data underlying the findings in their manuscript fully available?

Reviewer #1: Yes

Reviewer #2: Yes

4. Is the manuscript presented in an intelligible fashion and written in standard English?

Reviewer #1: Yes

Reviewer #2: Yes

**Reviewer #1: ** This manuscript explores the physiological and metabolic responses of O. maya offspring to heatwave conditions, focusing on oxidative stress, mitochondrial function, and survival. The findings shed light on the species' potential vulnerability and adaptive capacity under climate change scenarios. The study is timely and relevant, given ongoing shifts in environmental temperatures across species’ ranges. However, while the manuscript contributes valuable data, particularly for conservation and developmental physiology, several issues need to be addressed before it is suitable for publication in PLOS ONE.

1. The abstract states that energy was reallocated toward stress defence mechanisms at 30°C, resulting in impaired development, smaller hatchling size, and reduced survival. However, this conclusion appears to overreach based on the presented data. While changes in morphology, metabolic rate, and glutathione (GSH) levels may suggest physiological stress, they are not sufficient on their own to confirm a definitive energy reallocation mechanism. Stronger support would require direct measurements of energy budgeting across physiological systems (e.g., immune response, oxidative damage markers, stress hormones). Therefore, this interpretation should be moderated or explicitly framed as a hypothesis rather than a confirmed mechanism.

2. The design includes three incubation temperatures (26, 29, and 32°C), a standard approach. However, it is unclear whether these temperatures reflect natural nest variation. Additional justification for the chosen values (e.g. field-measured nest temperatures) is warranted.

3. RMR of embryos and hatchlings was measured and compared only at their respective acclimation temperatures, rather than being assessed under the same conditions (e.g., at 24°C, 26°C, and 30°C across all groups). The absence of temperature as a controlled variable may limit the comparability of the results.

4. The sample sizes for hatchling respiration experiments were small. RMR, HMR, and LMR data were based on only 5–6 individuals, while mitochondrial respiration measurements were based on just 2–3 samples, with considerable variability. Conclusions drawn from such small sample sizes must be treated with caution. The high variability in these data undermines confidence in statistical significance or biological relevance.

5. The statistical framework appears generally appropriate, but effect sizes and confidence intervals should be provided alongside p-values to give more context to the biological significance of observed differences.

**Reviewer #2:**  The authors' study the embryonic sensitivity of octopus when exposed to species specific heatwaves and through this the experimental design allowed authors to carefully work at the mechanisms during development including mitochondrial dysfunction, ROS and hatching dynamics. This is an important study and is crucial for the understanding of the field. We need data on how heatwaves impact all of life processes. Having said this the manuscript is written well for most part, just a few additional clarifications (please see my appended pdf with comments). In your data analysis section it would be helpful if you could mention which data did not conform to normality and as a result you had to transform them. Such specific information is required within the main text of your current ms. Also suggested some specific bits of information needed within other sections.

**Do you want your identity to be public for this peer review?** For information about this choice, including consent withdrawal, please see our Privacy Policy

Reviewer #1: No

Reviewer #2: No

---

## [Author Response · Author response to Decision Letter 1]

27 Jun 2025

We thank both reviewers for their constructive feedback and insightful suggestions, which have significantly strengthened our manuscript. We hope that the revisions and clarifications provided here, and in the revised manuscript, adequately address all concerns.

Reviewer #1: This manuscript explores the physiological and metabolic responses of O. maya offspring to heatwave conditions, focusing on oxidative stress, mitochondrial function, and survival. The findings shed light on the species' potential vulnerability and adaptive capacity under climate change scenarios. The study is timely and relevant, given ongoing shifts in environmental temperatures across species’ ranges. However, while the manuscript contributes valuable data, particularly for conservation and developmental physiology, several issues need to be addressed before it is suitable for publication in PLOS ONE.

1. The abstract states that energy was reallocated toward stress defence mechanisms at 30°C, resulting in impaired development, smaller hatchling size, and reduced survival. However, this conclusion appears to overreach based on the presented data. While changes in morphology, metabolic rate, and glutathione (GSH) levels may suggest physiological stress, they are not sufficient on their own to confirm a definitive energy reallocation mechanism. Stronger support would require direct measurements of energy budgeting across physiological systems (e.g., immune response, oxidative damage markers, stress hormones). Therefore, this interpretation should be moderated or explicitly framed as a hypothesis rather than a confirmed mechanism.

RESPONSE: We added “We hypothesized that” energy reallocation… to moderate the findings (L32).

2. The design includes three incubation temperatures (26, 29, and 32°C), a standard approach. However, it is unclear whether these temperatures reflect natural nest variation. Additional justification for the chosen values (e.g. field-measured nest temperatures) is warranted.

RESPONSE: Thank you for your observation. We would like to clarify that the study temperatures were 24, 26, and 30°C, not 26, 29, and 32°C. These values were selected to represent an optimal condition (24°C), an intermediate temperature (26°C), and a high temperature (30°C), the latter reflecting conditions recorded during marine heatwaves in the region.

As stated in the Introduction, the optimal reproductive temperature range for Octopus maya is narrowly defined between 24°C and 25°C. In field monitoring along the Yucatán coast at 10.5 m depth, we have documented bottom temperatures exceeding 28°C for durations of up to three months (see Fig. 1). Therefore, the selected temperatures were chosen to simulate both optimal and thermally stressful conditions observed or anticipated in the species’ natural habitat.

To improve clarity, we have revised the Introduction and Materials and Methods sections. We now explicitly describe 24°C, 26°C, and 30°C as optimal, intermediate, and high temperatures, respectively (L172-174). Additionally, we added the phrase “(reaching temperatures of 30°C)” to better contextualize the ecological relevance of the high-temperature treatment (L120).

3. RMR of embryos and hatchlings was measured and compared only at their respective acclimation temperatures, rather than being assessed under the same conditions (e.g., at 24°C, 26°C, and 30°C across all groups). The absence of temperature as a controlled variable may limit the comparability of the results.

RESPONSE: Thanks for your observation, In the original sentence, we did not clearly state that the observed differences in RMR were derived from statistical comparisons among incubation temperatures at each developmental stage. To improve clarity, we have revised the paragraph in the Results section to first state the significant interaction between developmental stage and incubation temperature, followed by post hoc comparisons between specific temperatures within each stage. This revised structure explicitly outlines which temperature groups differ and at which stage of development. We have also specified the reference group in each pairwise comparison. We hope this modification resolves the ambiguity you pointed out.

4. The sample sizes for hatchling respiration experiments were small. RMR, HMR, and LMR data were based on only 5–6 individuals, while mitochondrial respiration measurements were based on just 2–3 samples, with considerable variability. Conclusions drawn from such small sample sizes must be treated with caution. The high variability in these data undermines confidence in statistical significance or biological relevance.

RESPONSE: We appreciate the reviewer’s concern regarding the small sample sizes in our hatchling respiration (n = 5–6) and mitochondrial assays (n = 2–3). To enhance transparency, we have supplemented all p-values with partial η² and 95 % confidence intervals, and provided a complete breakdown of means, standard deviations, effect sizes, and confidence intervals for each group in Supplementary Table S1.

Importantly, all animals used conformed to the Comisión de Ética Académica y Responsabilidad Científica (CEARC) guidelines at the Facultad de Ciencias, UNAM, which emphasize the “3Rs” principle, as disclosed in the Ethics statement. In accordance with CEARC’s mandate to reduce animal use to the minimum necessary for statistically valid results, we deliberately limited our sample sizes while still ensuring robust, reproducible analyses.

These additions allow readers to fully assess the precision and magnitude of our findings, and demonstrate our commitment to both ethical and scientific rigor.

5. The statistical framework appears generally appropriate, but effect sizes and confidence intervals should be provided alongside p-values to give more context to the biological significance of observed differences.

RESPONSE: We have included effect sizes and confidence intervals alongside (univariate) p-values. In addition, we have included detailed statistics for these assays to S1 Table (Supplementary Materials), so that readers can directly assess means, SD, effect sizes, and confidence intervals for each group.

Reviewer #2: The authors' study the embryonic sensitivity of octopus when exposed to species specific heatwaves and through this the experimental design allowed authors to carefully work at the mechanisms during development including mitochondrial dysfunction, ROS and hatching dynamics. This is an important study and is crucial for the understanding of the field. We need data on how heatwaves impact all of life processes. Having said this the manuscript is written well for most part, just a few additional clarifications (please see my appended pdf with comments). In your data analysis section it would be helpful if you could mention which data did not conform to normality and as a result you had to transform them. Such specific information is required within the main text of your current ms. Also suggested some specific bits of information needed within other sections.

RESPONSE: We now specified the data not meeting normality (L401).

Reviewer 2 appended pdf with comments

Note: Reviewer 2 provided detailed feedback in an annotated PDF. Please note that LINE numbers referenced in their comments correspond to the original manuscript draft, whereas LINE numbers cited in our responses (enclosed in parentheses) refer to the revised manuscript version.

• L33-34. do we need such detail in the abstract? could this be simplified for the lay reader?

o RESPONSE: The details were removed (L33).

• L46. are detrimental to (consider rephrasing this)

o RESPONSE: suggestions accepted (L46)

• L54. With continued climate change ?

o RESPONSE: Yes, we added the word continued as suggested (L54).

• L60-64. This is essential information, but do break the sentences down otherwise the sentence structure is lost- rephrase/restructure this.

o RESPONSE: it was broken down in two sentences (L60-63)

• Did you have any specific predictions, directions of those predictions alongside the hypotheses? Stating that will strengthen this portion of your work

o RESPONSE: Great observation. We have added specific predictions (L123-130).

• are the animals age matched or how do you control for the age of the adults in your catch, did you control for this?

o RESPONSE: Yes, the animals used in this study were age-controlled. Octopus maya has a lifespan of approximately one year (Rosas 2024, doi: 10.1016/B978-0-12-820639-3.00009-1), and the individuals selected for this experiment were of similar body size (400–700 g), as detailed in the Methods section. This size range corresponds to the mature reproductive phase, given that sexual maturation typically occurs at ~334–342 g in males and ~333–335 g in females (Ávila-Poveda et al. 2016, doi: 10.1080/13235818.2015.1072912). Furthermore, all specimens were collected using the gareteo fishing method, which selectively captures sexually mature animals that are actively migrating to reproductive zones, prior to spawning and denning behavior. Additionally, a conditioning period was implemented to confirm the functional maturity of all individuals. Notably, O. maya populations at the Sisal location exhibit two main reproductive cohorts per year, which tend to synchronize in terms of reproductive timing (Ávila-Poveda et al. 2016, doi: 10.1080/13235818.2015.1072912). Therefore, both the biological characteristics and the local population dynamics support that the animals used were of comparable age and reproductive status.

• L170. After this?

o The word “after” was replaced by “Subsequently” for clarity (L172).

• L281-282. could you provide an estimate of mortality? absolute or % percentage mortality?

o RESPONSE: Yes, as reported in the Results section, survival rates for juveniles were 92.5% at 24 °C, 82.5% at 26 °C, and 52.5% at 30 °C, corresponding to mortality rates of 7.5%, 17.5%, and 47.5%, respectively (491-492).

• L424 could you give the figure details? like a number

o RESPONSE: Thank you for pointing this out. We realized that the confusion regarding the figure reference stemmed from missing details on the construction of the PCO plot. Specifically, the figure includes embedded visual representations of centroid positions and relative distances to aid interpretation of group clustering. This clarification has now been added to the Materials and Methods section (407-409).

• L438. It would be better to state the effect rather than using ambiguous language- state as is.

o RESPONSE: It was stated as is (L443-445).

• L439-440. this does not make sense? can you rephrase this please?

o RESPONSE: It was rephrased (L443-445).

• L440-441. i can understand the use of abbreviations here- but as a reader i found it harder to follow and appreciate the importance of your results. Is there a way you could consider making it easier?

o RESPONSE: Sure, for easier readability we spell them out even if they were already spelled out in Materials and Methods section (L443-448).

• L444-445. just by stating that it is statistically different means little, can you provide better context here- of course if you change temperature during embryonic development it is going to induce changes in traits, but how is it important? You could state the relationship of these morphological differences, directionality, what percentage it changed and how did it compare across the traits you considered

o RESPONSE: It was rewritten stating context (L447-449).

• L447. is this compared to some baseline? all three incubation temperatures showed a difference then this difference is compared to what?

o RESPONSE: It was rewritten (L451-459).

• L458. all incubation temperatures, need to be specific here?

o RESPONSE: It was rewritten for enhanced clarity (L466-467).

• L555-557. could this also be linked to terminal investment of the parent as heavier embryos (in general, also seen in humans where heavier foetuses- just as a example) survive well, selection on mothers to improve chances of embryo survival, best of the bad situation etc..

o RESPONSE: Thank you for this insightful observation. While the idea of terminal investment is interesting and relevant in other systems, we believe that the differences in embryo morphometry observed in our study are more likely attributed to direct metabolic alterations during embryonic development, as discussed throughout the manuscript, rather than a transgenerational or maternal investment effect. All females used in this study were maintained under optimal thermal conditions (24 °C), minimizing the likelihood of temperature-induced maternal effects. We agree that testing for terminal investment or adaptive maternal provisioning would require experimental manipulation of maternal environmental conditions. In fact, we previously conducted a study exploring possible transgenerational plasticity by exposing O. maya females to different temperatures (reference cited in L636-638 and L653–658), which may provide a more appropriate framework for evaluating this hypothesis.

• L558-560. sentence structure- greater?

o RESPONSE: The sentence was rewritten for better clarity (L569-572)

• L564. main what?

o RESPONSE: The sentence was rewritten for better clarity (L575).

• L577-579. a way this can be provisioned may be that the gametes at the fertilisation are primed to allocate key aspects of embryonic development (alluded to here: https://doi.org/10.1098/rsos.23142

o RESPONSE: Thanks for the suggestion. We have incorporated the idea and reference (L587-590).

---

## [Decision Letter · Decision Letter 1]

5 Aug 2025

Can octopus embryos and juveniles contend with heatwaves?

PONE-D-25-14615R1

Dear Dr. Rosas,

We’re pleased to inform you that your manuscript has been judged scientifically suitable for publication and will be formally accepted for publication once it meets all outstanding technical requirements.

Kind regards,

Juan J Loor

Academic Editor

PLOS ONE

Additional Editor Comments (optional):

Reviewers' comments:

Reviewer's Responses to Questions

**Comments to the Author**

Reviewer #2: All comments have been addressed

2. Is the manuscript technically sound, and do the data support the conclusions?

Reviewer #2: Yes

3. Has the statistical analysis been performed appropriately and rigorously?

Reviewer #2: Yes

4. Have the authors made all data underlying the findings in their manuscript fully available?

Reviewer #2: Yes

5. Is the manuscript presented in an intelligible fashion and written in standard English?

Reviewer #2: Yes

Reviewer #2: Thank you for clarifying the points raised through the peer-review and i am delighted to read the changes and see how it has improved your ms. I do not have any further comments at this stage.

Best wishes

**Do you want your identity to be public for this peer review?** For information about this choice, including consent withdrawal, please see our Privacy Policy

Reviewer #2: No

---

## [Editor Report · Acceptance letter]

PONE-D-25-14615R1

PLOS ONE

Dear Dr. Rosas,

I'm pleased to inform you that your manuscript has been deemed suitable for publication in PLOS ONE. Congratulations! Your manuscript is now being handed over to our production team.

Kind regards,

on behalf of

Dr. Juan J Loor

Academic Editor

PLOS ONE